# Perioperative Evaluation of the Physical Quality of Life of Patients with Non-Small Cell Lung Cancer: A Prospective Study

**DOI:** 10.3390/cancers16081527

**Published:** 2024-04-17

**Authors:** Ryuta Fukai, Tomoki Nishida, Hideyasu Sugimoto, Makoto Hibino, Shigeto Horiuchi, Tetsuri Kondo, Shinichi Teshima, Masahiro Hirata, Keiko Asou, Etsuko Shimizu, Yuichi Saito, Yukinori Sakao

**Affiliations:** 1Department of General Thoracic Surgery, Shonan Kamakura General Hospital, 1370-1, Okamoto, Kamakura 247-8533, Kanagawa, Japan; tmk-n.322@nifty.com; 2Department of Respiratory Medicine, Saiseikai Yokohamashi Nanbu Hospital, 3-2-10, Konandai, Konan-ku, Yokohama 234-0054, Kanagawa, Japan; hideyasu-sugimoto@live.jp; 3Department of Respiratory Medicine, Shonan Fujisawa Tokushukai Hospital, 1-5-1, Tsujidokandai, Fujisawa 251-0041, Kanagawa, Japan; m-hibino@ctmc.jp (M.H.); shigeto.horiuchi@ctmc.jp (S.H.); tetsuri@ctmc.jp (T.K.); 4Department of Pathology, Shonan Kamakura General Hospital, 1370-1, Okamoto, Kamakura 247-8533, Kanagawa, Japan; steshima@shonankamakura.or.jp; 5Center for Clinical and Translational Science, Shonan Kamakura General Hospital, 1370-1, Okamoto, Kamakura 247-8533, Kanagawa, Japan; m_hirata@shonankamakura.or.jp; 6Center for Clinical Research, Shonan Kamakura General Hospital, 1370-1, Okamoto, Kamakura 247-8533, Kanagawa, Japan; k_asou2@shonankamakura.or.jp; 7Clinical Research Center, Shonan Fujisawa Tokushukai Hospital, 1-5-1, Tsujidokandai, Fujisawa 251-0041, Kanagawa, Japan; etsuko.shimizu@tokushukai.jp; 8Department of Surgery, Teikyo University School of Medicine, 2-11-1 Kaga, Itabashi-ku, Tokyo 173-0003, Japan; yuichi.saito@med.teikyo-u.ac.jp (Y.S.); ysakao@med.teikyo-u.ac.jp (Y.S.)

**Keywords:** physical quality of life, lung cancer, surgery, smoking status, performance status, living conditions, Charlson comorbidity index

## Abstract

**Simple Summary:**

Surgery is the most effective treatment for early-stage lung cancer, but it poses a heavy physical burden. Accordingly, understanding the perioperative daily life conditions of patients is important to maintain their health status and to provide appropriate treatment. We performed a prospective study to examine the socioclinical factors associated with the physical quality of life of patients who underwent surgery for lung cancer at Shonan Kamakura General Hospital, Kanagawa, Japan. In the preoperative setting, living alone and lower performance status were independently associated with worse physical quality of life. In the postoperative setting at 6 months, later smoking cessation, lower performance status, living alone, and higher comorbid burden were independently associated with worse physical quality of life. In order to maintain quality of life and provide enough treatment, perioperative management should include taking care of the patient’s physical condition, lifestyle, smoking, and comorbid status.

**Abstract:**

Surgery is the most effective treatment for early-stage lung cancer; however, it poses a higher physical burden than other treatment options. Therefore, understanding the perioperative course of patients is important. Using the Short Form Health Survey 36, we prospectively measured the physical quality of life of patients who underwent anatomical pulmonary resection for non-small cell lung cancer at Shonan Kamakura General Hospital, Kanagawa, Japan (n = 87). In the preoperative setting, patients who had lower performance status and lived alone had significantly worse physical quality of life scores on multivariate analysis (regression coefficient (95% confidence interval), −9.37 (−13.43–−5.32) and −10.22 (−13.74–−7.40), respectively, *p* < 0.0001 for both). At 6 months postoperatively, patients who stopped smoking within 1 year preoperatively (stopped smoking within 1 year vs. remote or never smokers, 41.0 ± 10.5 vs. 48.6 ± 7.2, *p* = 0.002), had lower performance status (0 vs. 1–2, 49.3 ± 6.6 vs. 38.6 ± 9.6, *p* < 0.0001), lived alone (living alone vs. living with somebody, 41.6 ± 9.7 vs. 48.1 ± 7.9, *p* = 0.021), and had higher comorbid burden (Charlson comorbidity index <3 vs. ≥3, 48.2 ± 6.9 vs. 39.1 ± 14.7, *p* = 0.003) had significantly worse physical quality of life scores on univariate analysis. More recent smoking (regression coefficient (95% confidence interval), −4.90 (−8.78–1.0), *p* = 0.014), lower performance status (8.90 (5.10–12.70), *p* < 0.0001), living alone (5.76 (1.39–10.13), *p* = 0.01), and higher comorbid burden (−6.94 (−11.78–−2.10), *p* = 0.006) were significant independent predictors of worse postoperative physical quality of life on multivariate analysis. Therefore, patients with these conditions might need additional support to maintain their physical condition after anatomical lung cancer surgery.

## 1. Introduction

Surgery is considered the most effective and best curative option for patients with early-stage lung cancer, but it poses a higher physical burden than other treatment modalities for lung cancer. Lung cancer is a disease of the elderly [1] and the leading cause of cancer-related mortality worldwide [2]. In fact, septuagenarians comprise the most common surgical population in Japan [3]. As aging of the population progresses, evaluation of the influence of surgery on patients has become more meaningful, because compared with younger patients, elderly patients with lung cancer have more comorbidities and are more vulnerable [4].

In some countries, there has been a shift from big families to nuclear families in recent decades [5,6,7]. This change in family structure has increased the number of elderly people living alone. Under these circumstances, detailed understanding of the perioperative course, especially the daily life, of patients with lung cancer has become more important.

Evaluation of health-related quality of life (HR-QOL) has been reported to be a method for the assessment of the postoperative condition of these patients. Several reports have confirmed that the postoperative decline in QOL is associated with survival after lung cancer surgery and the course of treatment [8,9]. We previously reported that the perioperative mental QOL of patients with lung cancer was significantly associated with smoking status, living alone, and the Charlson comorbidity index (CCI) [10]. This study aimed to investigate the socioclinical factors associated with the perioperative physical QOL (P-QOL) using the Short Form Health Survey 36 (SF-36) to understand the influence of surgery and adjuvant therapies on the P-QOL of patients, especially in daily life.

## 2. Patients and Study Methods

Two thoracic surgeons (R.F. and T.N.) performed the surgeries of this study. These surgeons worked as full-time employees at Shonan Kamakura General Hospital (SKGH), and part-time employees at Shonan Fujisawa Tokushukai Hospital (SFTH), which is about 10 km from SKGH due to the lack of a thoracic surgeon in SFTH. Of the 108 patients who underwent surgery for non-small cell lung cancer (NSCLC) at our institution between April 2015 and November 2017, 100 patients provided written informed consent to participate in this study during the preoperative visit at the time of consenting for surgery. The participants were asked to fill out the questionnaire within 1 month of surgery at preoperative visit and at 1, 3, 6, and 12 months postoperatively as an outpatient at SKGH or SFTH by thoracic surgeons. Patients who underwent partial resection (n = 8), repeat lobectomy (n = 1), lobectomy with infiltrated rib resection (n = 1), and exploratory thoracoscopic surgery (n = 1) were excluded because the surgical stress of these procedures is fundamentally different from that of initial lobectomy or segmentectomy. Patients facing difficulty in completing their questionnaires postoperatively because of major complications (n = 2, cerebral infarction and severe brain edema, respectively) were also excluded. Finally, 87 patients who underwent anatomical pulmonary resection for NSCLC were analyzed; 74 patients (85%) underwent surgery thoracoscopically and 13 (15%) by thoracotomy. The patient’s data, including postoperative adjuvant therapies, were collected through the medical record review.

### 2.1. Quality of Life Assessment

QOL was assessed using the Japanese version of SF-36 [11]. The SF-36, a self-rated questionnaire that comprises 36 items grouped into eight scales, including physical and mental health, is used to assess eight QOL dimensions [12]. As the main focus of the study was to understand the effect of surgery and adjuvant therapies on the P-QOL of the patients, especially in daily life, four domains from the eight SF-36 subscales were selected as elements of physical health. In this study, we evaluated only physical health, which comprises the following four subscales: physical function (PF), role-physical (RP), bodily pain (BP), and general health (GH) (Appendix A).

The raw scores were standardized and ranged from 0 to 100, where 0 represented the worst stage of health and 100 represented the best possible. The national standard level (NSL) with a confidence interval of 95% was used to consider the physical health status. The reliability and validity of the SF-36 questionnaire have been confirmed in international cancer studies [13,14,15,16].

### 2.2. Smoking Status

During the first visit, the surgeons instructed all patients to stop smoking; thereafter, we checked with the patients and their family whether they had stopped smoking completely at a follow-up visit. We did not use any tools or support for stopping smoking. If the patient could not stop smoking, we refused their request for surgery. Smoking status was classified into two groups, including stopped smoking within 1 year and remote or never smokers (i.e., never smoked or stopped smoking more than 1 year preoperatively). All study participants stopped smoking at least 2 weeks preoperatively.

### 2.3. Performance Status

To evaluate the preoperative physical activity of the patients, we used the Eastern Cooperative Oncology Group (ECOG) performance status (PS), which includes the following scores: 0, fully active; 1, restricted in physically strenuous activity but ambulatory and able to carry out work of a light or sedentary nature; 2, ambulatory and capable of self-care but unable to carry out any work activities; 3, capable of only limited self-care; 4, completely disabled; and 5, dead [17]. We confirmed the preoperative PS of each patient using the patients’ medical records.

### 2.4. Living Conditions

We classified the living conditions of patients into two groups, living alone and with somebody.

### 2.5. Charlson Comorbidity Index

We evaluated the CCI of each patient based on the preoperative comorbid status. Patients were considered to have a comorbidity if a listed disease was confirmed on the medical records or if the patient received treatment for it. We used the modified CCI, as proposed by Birim et al. [18].

We prospectively investigated the progress of perioperative P-QOL as well as the relationship between preoperative and 6-month postoperative P-QOL and patient characteristics, including age, sex, smoking status, ECOG PS, living conditions, and CCI. The scores at 6 months postoperatively were chosen as the postoperative P-QOL because most of the postoperative scores of the four P-QOL subscales plateaued at 3–6 months postoperatively, which is similar to other studies [19,20,21]. Patient data were extracted from the medical records. This study was approved by our institutional review board.

### 2.6. Statistical Analysis

In accordance with the SF-36 procedures, the scores were converted to a linear scale that ranged from 0 to 100 for each patient. For each of the four subscales, lower scores represented greater symptom burden. For all patients, the mean score of the four subscales was calculated and used to evaluate the relationship between socioclinical factors and P-QOL. All statistical analyses were performed using EZR version 1.55 (free statistical software), which is a modified version of the R commander and was designed to add statistical functions that are frequently used in biostatistics [22].

The paired t-test was used to compare the preoperative and postoperative mean P-QOL scores on all four subscales. The Student’s t-test was used to compare preoperative and postoperative P-QOL scores between two groups, which were divided based on each clinical factor (age, sex, smoking status, PS, living conditions, and CCI). Multivariate analysis was used for the factors with significant differences on univariate analysis. Multiple regression analysis was used for multivariate analysis.

## 3. Results

Overall, 87 patients participated in this study and the survey completion percentage (the number of omitted responses) to all items of the QOL questionnaire was 100% (0) preoperatively and 91% (21), 96% (12), 89% (36), and 89% (36) at 1, 3, 6, and 12 months postoperatively, respectively. The response rate of the surveys was 100% preoperatively and 94%, 97%, 90%, and 90%, at 1, 3, 6, and 12 months postoperatively, respectively. Of the 87 patients, 64 (74%) underwent surgery alone, while 23 (26%) underwent adjuvant therapy within 1 year postoperatively, including platinum-based chemotherapy (n = 11, 48%), oral uracil-tegafur (n = 10, 44%), radiotherapy (n = 1, 4%), and afatinib (n = 1, 4%). Of the 11 patients (13%) who had lung cancer recurrence within 1 year postoperatively, only 1 (1%) died because of lung cancer. The sociodemographic and clinical characteristics of the patients are presented in Table 1.

### 3.1. Baseline P-QOL Subscales

All baseline (preoperative) P-QOL scores were equal to the Japanese NSL (50%) in the four subscales (Figure 1). On univariate analyses, the mean value of the four P-QOL subscales significantly differed according to age (*p* = 0.022), smoking status (*p* = 0.025), living conditions (*p* < 0.001), and ECOG PS (*p* < 0.0001) (Table 2), but not significantly by sex and CCI. We conducted multivariate analysis for these four factors showing significant differences on univariate analysis. Living alone and lower PS (≥1) were found to be significantly associated with worse preoperative P-QOL scores on multivariate analysis (Table 3). Specifically, the mean value of P-QOL scores on all four subscales of the patients living alone with lower PS (≥1) was 33.4, and that of those living with somebody with good PS (0) was 54.0.

### 3.2. Postoperative P-QOL Evaluation

The P-QOL scores on all four subscales significantly decreased at 1 month postoperatively compared with the preoperative scores (*p* < 0.0001). Further, they almost plateaued 6 months postoperatively. The P-QOL scores on BP recovered to baseline at 3 months postoperatively, but the PF (*p* = 0.002) and RP (*p* = 0.028) scores significantly decreased even at 1 year postoperatively compared with the preoperative scores (Figure 1).

### 3.3. Predictors of Postoperative P-QOL

We compared the patient characteristics and postoperative P-QOL scores, which had plateaued at 6 months postoperatively. On univariate analysis (Table 4), the mean P-QOL on the four subscales at 6 months postoperatively significantly differed according to smoking status (stopped smoking within 1 year vs. remote or never smokers, 41.0 ± 10.5 vs. 48.6 ± 7.2, *p* = 0.002), PS (0 vs. 1–2, 49.3 ± 6.6 vs. 38.6 ± 9.6, *p* < 0.0001), living conditions (living alone vs. living with somebody, 41.6 ± 9.7 vs. 48.1 ± 7.9, *p* = 0.02), and CCI (<3 vs. ≥3, 48.2 ± 6.9 vs. 39.1 ± 14.7, *p* = 0.003). Even on multivariate analysis, all four factors significantly affected the P-QOL at 6 months postoperatively (Table 5).

## 4. Discussion

Surgery is considered the best treatment for patients with early-stage NSCLC, but its invasiveness had a negative impact on patient HR-QOL. This study revealed that the P-QOL scores on all four subscales significantly decreased at 1 month postoperatively compared with baseline, and they plateaued at 6 months postoperatively. The P-QOL scores on PF and RP did not recover even at 1 year postoperatively, even though majority (85%) of the surgeries were performed thoracoscopically. Preoperatively, the patients who lived alone and with lower PS (≥1) had significantly lower P-QOL scores in multivariate analysis. Moreover, postoperatively, multivariate analysis revealed that later smoking cessation (i.e., within 1 year preoperatively), lower PS (≥1), living alone, and more comorbid status (CCI ≥ 3) were independently associated with significantly lower P-QOL at 6 months postoperatively.

PS is an important factor in determining the QOL, choice of treatment, and prognosis of patients with cancer. A better PS suggests a better prognosis [23]. In patients with lung cancer, poor PS is a known negative prognostic factor [24,25]. Conversely, only a few articles have reported the association between PS and perioperative P-QOL in patients with lung cancer. Our results revealed that lower PS (≥1) was independently associated with significantly lower preoperative and postoperative P-QOL. In a randomized controlled trial, Liu Z et al. reported that a short-term (2 weeks) multimodal rehabilitation strategy in the perioperative period improved the functional capacity of patients who underwent thoracoscopic lobectomy for lung cancer [26]. For patients with lung cancer and PS ≥ 1, we may consider perioperative rehabilitation in order to maintain their QOL and provide curative treatment.

Living alone affects the HR-QOL of patients with multiple comorbidities [27], and several authors have found that living conditions may affect the survival of patients with cancer [28,29,30]. In this study, we found that the patients who lived alone had significantly worse preoperative and postoperative P-QOL scores. We think that living alone, especially for the elderly, limits support from family members, which then makes remaining motivated to overcome their disease and receive necessary medical treatment more challenging. Moreover, Cheng et al. reported that older adults who lived alone were at a high risk of developing sarcopenia and probable sarcopenia [31]. Sarcopenia is defined as a progressive loss of muscle mass and reduced muscle strength and functional ability, which can lead to worse P-QOL. The factor of “living alone” should be considered as one of the risks to P-QOL. Living conditions of patients during perioperative period should be accounted for to avoid QOL deteriorating by cooperating with nurses and social workers.

Smoking has certain negative impacts on human QOL [32,33] and is a risk factor for the development of various cancers, including lung cancer [34,35]. Accordingly, our results revealed that the preoperative and postoperative P-QOL scores were relatively worse in patients who stopped smoking within 1 year. The findings might not be generalized, as not all centers/countries deny surgery to patients who continue to smoke; thus, smoking cessation in the postoperative period is a confounding variable. Preservation of QOL is important for providing necessary treatment and improving the treatment outcomes or prognosis of patients. Goldenberg et al. demonstrated that smoking cessation significantly improved QOL and that compared with nonsmokers, smokers consistently reported lower physical domain scores [32]. In addition, Hays et al. noted that a longer period of smoking cessation produced better HR-QOL [36]. They also found that compared with placebo, the nonnicotine medications varenicline and SR bupropion for the treatment of tobacco use and dependence led to improved HR-QOL. These results suggested that smoking cessation treatment should be considered for patients who find it difficult to stop smoking preoperatively and to prevent postoperative smoking relapse. Recent quitters (i.e., stopped smoking within 4 weeks of surgery) appeared to have an increased incidence of pulmonary complications [37], although no pulmonary complications were observed in six patients who had stopped smoking within 4 weeks of surgery.

The CCI, which was developed by Charlson et al. in 1987 [38], had been strongly associated with a relatively high risk of surgery in patients with NSCLC. Moreover, in two large-scale phase III trials of the National Cancer Institute of Canada, an unfavorable CCI score was associated with a poor prognosis [39]. In patients with prostate cancer, CCI was reported to be useful, mainly for predicting long-term QOL and PF scores [40]. In this study on patients with NSCLC, a CCI of ≥3 was found to lead to worse postoperative P-QOL. However, many clinical trials frequently exclude patients with the common comorbidities and do not perfectly reflect the characteristics of patients who are routinely treated for NSCLC. Therefore, we considered that compared with PS assessment by physicians, the SF-36 patient self-assessment of QOL score would more accurately reflect the real health status of the patients [41]. Compared with the conventional PS analysis, both the CCI and QOL score may be more accurate indicators of the health status of patients with NSCLC and might help in the decision on the best and individualized therapeutic option.

Our study had some limitations. First, it was conducted at a single institution on a small scale. Second, we could not obtain complete responses to the self-rated questionnaire from the patients. Third, we did not consider the influence of postoperative adjuvant therapy on P-QOL, although the majority of the patients underwent only surgery. Of the 23 patients who underwent adjuvant therapies, 10 patients (44%) underwent oral uracil-tegafur. It has been reported that overall HR-QOL did not deteriorate during adjuvant chemotherapy with oral uracil-tegafur plus leucovorin in patients with colorectal cancer [42]. Among ten patients, only one patient experienced Grade 3 pancytopenia at 10 months after the beginning of uracil-tegafur, while nine patients had only Grade 1 toxicities. These findings suggest that oral uracil-tegafur has less negative impact than platinum-based systemin chemotherapy, which has been reported to have severe side effects such as nephrotoxicity, hematological toxicity, gastrointestinal toxicity, and neurotoxicity [43]. Therefore, we believe that the total negative impact of adjuvant therapies on the P-QOL of the patients was relatively mild. Fourth, we evaluated only four physical domains of the eight subscales of SF-36. During SF-36 validity testing, PF and RP had the most pure physical health interpretation [44]. Moreover, during SF-36 reliability testing, all eight subscales showed high intrinsic consistency [12]. From these results, we believe the method to evaluate P-QOL used in this study is appropriate.

Despite these limitations, our results are clinically meaningful, because the evaluation of postoperative status was performed by the patients themselves, and the identified independent predictors of worse postoperative P-QOL can be confirmed preoperatively. Therefore, preoperative use of the SF-36 can potentially identify patient factors that increase the likelihood of lower postoperative P-QOL.

## 5. Conclusions

Living alone and lower PS (≥1) were independently associated with significantly worse P-QOL in preoperative settings, while later smoking cessation, PS ≥1, living alone, and higher comorbid burden (CCI ≥ 3) were independently associated with significantly worse P-QOL at 6 months postoperatively. Even after 1 year and with the use of less invasive procedures in the majority of cases, patients with lung cancer experienced a reduced P-QOL in the postoperative period, which might be due to the effects of surgery and other adjuvant therapies. Further studies are warranted to differentiate the effect of surgery from that of adjuvant therapies to P-QOL. To maintain the P-QOL and provide more fruitful medical treatment of this population, attention should be paid to their physical condition, lifestyle, smoking, and comorbid status and necessary measures should be taken in accordance with the situation during perioperative surgical management.

## Figures and Tables

**Figure 1 cancers-16-01527-f001:**
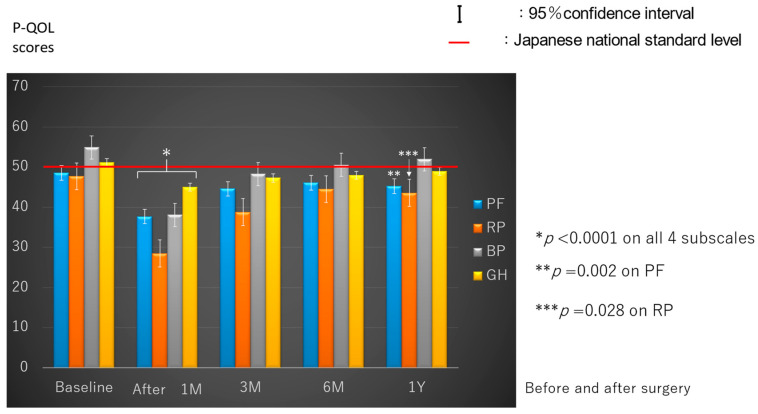
Perioperative progress on the four subscales of physical quality of life. PF, physical function; RP, role-physical; BP, bodily pain; GH, general health.

**Table 1 cancers-16-01527-t001:** Sociodemographic and clinical characteristics of patients (N = 87).

Characteristics	Number (%)
Age (years)	
Mean ± SD (range)	69.7 ± 8.5 (48–83)
Sex	
Male	41 (47)
Female	46 (53)
Performance status	
0	72 (83)
1–2	15 (17)
3–5	0 (0)
Smoking status	
Stopped within 1 year preoperatively	17 (20)
Remote or never smokers	70 (80)
Living conditions	
Living alone	10 (11)
Living with somebody	77 (89)
Charlson comorbidity index	
<3	77 (89)
≥3	10 (11)
Surgical approach	
Thoracoscopic	74 (85)
Thoracotomy	13 (15)
Procedure	
Segmentectomy	5 (6)
Lobectomy	79 (91)
Bilobectomy	2 (2)
Pneumonectomy	1 (1)
Histology	
Adenocarcinoma	73 (84)
Squamous cell carcinoma	12 (14)
Adenosquamous	1 (1)
Carcinoid	1(1)
Pathologic stage	
IA	35 (40)
IB	27 (31)
IIA	7 (8)
IIB	12 (14)
IIIA	6 (7)

SD, standard deviation.

**Table 2 cancers-16-01527-t002:** Association between clinical factors and preoperative physical quality of life on univariate analysis.

Variables	No. of	P-QOL Score	*p*-Value
	Patients (%)	(Mean ± SD)	
Age (years)			0.022
<70	35 (40)	53.0 ± 6.1	
≥70	52 (60)	46.5 ± 8.4	
Sex			0.25
Male	41 (47)	49.5 ± 9.0	
Female	46 (53)	51.5 ± 7.1	
Smoking status			0.025
Stopped within	17 (20)	46.7 ± 10.6	
1 year preoperatively			
Remote or never	70 (80)	51.5 ± 7.1	
smokers			
Performance status			<0.0001
0	72 (83)	52.8 ± 6.0	
1–2	15 (17)	41.8 ± 9.2	
Living conditions			<0.001
Living alone	10 (11)	42.0 ± 10.4	
Living with	77 (89)	51.7 ± 7.1	
somebody			
Charlson comorbidity index			0.47
<3	77 (89)	50.8 ± 7.3	
≥3	10 (11)	48.8 ± 12.9	

P-QOL, physical quality of life; SD, standard deviation.

**Table 3 cancers-16-01527-t003:** Association between clinical factors and preoperative physical quality of life on multivariate analysis.

Variables	Regression Coefficient (95% CI)	*p*-Value
Age	−0.88 (−3.80–2.05)	0.553
(≥70 years or not)		
Smoking status	2.47 (−0.96–5.90)	0.156
(Remote or never smokers or not)		
Living conditions	−9.37 (−13.43–−5.32)	<0.0001
(Living alone or not)		
Performance status	−10.22 (−13.74–−7.40)	<0.0001
(≥1 or not)		

CI, confidence interval.

**Table 4 cancers-16-01527-t004:** Association between clinical factors and postoperative physical quality of life scores on univariate analysis.

Variables	No. of	P-QOL Score	*p*-Value
	Patients (%)	(Mean ± SD)	
Age (years)			0.336
<70	35 (40)	47.4 ± 8.4	
≥70	52 (60)	46.5 ± 8.4	
Sex			0.43
Male	41 (47)	46.4 ± 10.0	
Female	46 (53)	47.9 ± 6.9	
Smoking status			0.002
Stopped within	17 (20)	41.0 ± 10.5	
1 year preoperatively			
Remote or never	70 (80)	48.6 ± 7.2	
smokers			
Performance status			<0.0001
0	72 (83)	49.3 ± 6.6	
1–2	15 (17)	38.6 ± 9.6	
Living conditions			0.021
Living alone	10 (11)	41.6 ± 9.7	
Living with	77 (89)	48.1 ± 7.9	
somebody			
Charlson comorbidity index			0.003
<3	77 (89)	48.2 ± 6.9	
≥3	10 (11)	39.1 ± 14.7	

P-QOL, physical quality of life; SD, standard deviation.

**Table 5 cancers-16-01527-t005:** Association between clinical factors and postoperative physical quality of life on multivariate analysis.

Variables	Regression Coefficient (95% CI)	*p*-Value
Smoking status	−4.90 (−8.78–1.0)	0.014
(Stopped smoking within 1 years or not)		
Performance status	8.90 (5.10–12.70)	<0.0001
(0 or not)		
Living conditions	5.76 (1.39–10.13)	0.01
(Living with somebody or not)		
Charlson comorbidity index	−6.94 (−11.78–−2.10)	0.006
(≥3 or not)		

CI, confidence interval.

## Data Availability

The data presented in this study are available on request from the corresponding author.

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
