# Peer review of "Perioperative Evaluation of the Physical Quality of Life of Patients with Non-Small Cell Lung Cancer: A Prospective Study"

_cancers, 2024, doi:10.3390/cancers16081527_

Round 1

Reviewer 1 Report

Comments and Suggestions for Authors

This study seeks to evaluate the impact of surgery on physical QOL of patients undergoing surgical resection for NSCLC both in the pre- and post-operative periods, and to determine patient characteristics that are associated with worse physical status.  The study is interesting in terms of how it provides longitudinal assessment of physical QOL overtime, but there are several limitations to the study. 

Firstly, the study does not add a lot of new information to the literature – it is not novel to report that older patients with more comorbidities and worse PS ultimately experience worse physical QOL pre or post-operatively.  It is, however, noteworthy, that some measures of physical QOL did not return to baseline even after 1 year, which would be important information for a patient to know when consenting for surgery. 

A major limitation to the study is that the authors do not indicate whether surgical treatment alone was curative, and what proportion of patients required additional therapies (e.g. adjuvant systemic therapy) in the post-operative period within 1 year.  Note is made that 29 patients ultimately had Stage II-III disease and would potentially be eligible for adjuvant therapies. This would potentially significantly affect P-QOL post-operatively.  At the minimum, the authors should include mention of what proportion of patients received surgical treatment alone vs. received additional therapies.  Additionally, it is suggested that the authors revise the analysis to exclude patients that underwent additional therapies in the post-operative period, so as to eliminate the potential confounding factor from the analysis and in order to ensure the analysis pertains to the impact of surgery alone on QOL.  As it stands, the authors conclude “that surgery had a negative impact on P-QOL of elderly patients with lung cancer” but this conclusion is mired by the fact that patients receiving adjuvant therapies were not excluded.

Another limitation is that this study does not include patients that continued to smoke in the post-operative period, limiting comparison to those that smoked within 1 year vs “others”, which is not clearly described, given that the authors state that those that continued smoking were denied surgery (which is not a universal standard in cancer care).

Additionally, the main comparison made in physical QOL scores is to the Japanese national standard level. The more important comparison is the comparison at the various post-operative time periods with the pre-operative baseline, and should be the main Figure of the paper. While this comparison is described in section 3.2, it should be the focus of the analysis.

Finally, this paper requires revision from an English proficiency perspective in that many terms are misleading, suboptimally worded, and there are grammatical errors throughout. Additional detailed feedback is provided below.

Simple Summary:

-              Line 32-33 - Suggest changing “sufficient treatment” to “appropriate treatment” or “curative treatment”

Abstract:

-              Line 42 - Suggest rewording “strongest physical load” to “highest physical risk” or “highest physical burden”.  Consider being more consistent in this terminology throughout the paper.

-              Line 43 - Rather than “prone to be affected by surgery” consider “at higher risk of complications” or “more likely to experience physical decline due to surgery”

-              Abstract must include details about the number of patients studied and numerical data regarding the results. For the main results stated in the abstract, numerical data with p-values should be presented. 

-              Condense the first 3 sentences of the abstract to allow for additional data to be included.

-              Significant revision of the grammatical structure is required.  E.g. rather than “patient who fulfill the aforementioned conditions”, consider  “patients with these conditions”. There should be no comma after “additional support” on line 53.

Introduction

-              Line 59 – 60 - 1st sentence – rather than “for early stage-lung cancer” consider “for patients with early stage lung cancer”

-              The 2nd paragraph of the Introduction can be significantly shortened for brevity.

-              Line 71  - again, the term “sufficient medical treatment” is suboptimal and should be revised to either “optimal treatment” or “curative intent treatment”.

-              Lines 76-78 need to be clarified. What changes in QOL are associated with survival and what does “progress of treatment” mean?

Methods

-              Line 91 - Rather than “Finally”, the better wording is “Ultimately”

-              Please add the percentages for all numbers provided e.g. 100 patients (%); 74 patients (%) undergoing thoracoscopic surgery and % undergoing thoracotomy as N (%).

-              In the methods, there needs to be more description about how the authors managed the fact that some patients were not given the survey to complete at certain time points – how did this factor into the evaluation?

-              Please describe where patients were consented? At clinic? At pre-operative visit? By phone?

-              What does “mode of living” mean? Please define categories. Note is made that this is defined in Section 2.5. Suggest including definitions up front for clarity

-              From the study methodology, it’s not clear why “the scores at 6 months after surgery were chosen”. The rest of the sentence makes it sound as though data were  collected at multiple post-operative time points but only the 6 month data were used. The methodology should include a description of the frequency of the post-operative survey data collection. (This is described in Section 2.1. The methods should be reorganized to include this information earlier).

-              Line 106  - Was the described outpatient consultation with the surgeon, anesthesia, or other?

-              More description of the 4 subscales of the SF-36 is required in the methods. What does “role-physical” include? What are the components that make up these subscales? Consider including the SF-36 as an Appendix.

-              What smoking cessation tools were used at the first visit? 2.3 simply states that patients were instructed to stop smoking. By whom? What supports were put in place? Was any information or tools provided?

-              Line 122 - How was it “confirmed adequately whether they stopped smoking completely”? Did a research assistant call? Was it by survey? Were patients asked at a follow-up visit?

-              Line 124-125 – what does “others” mean with respect to smoking cessation.

-              Line 133 – rather than “such as”, change to “including living alone and living with somebody” which is clearer.

-              The Methods section would benefit from reorganization to ensure clarity.  The 2nd paragraph of the Methods should move to just before the Statistical Analysis as it describes how the variables were used to assess the relationship with physical QOL.

-              When describing performance status, all 4 levels should be described, not just 1-3 and 5.

-              Line 104 – the Methods state that “each patient completed the questionnaire…” but in the results, the authors state that some patients weren’t given the questionnaire.  As such, it would be more clear to simply state “Patients were asked to complete the questionnaire …” and in the results, clarify how many patients were given the questionnaire.

-              Why did the authors choose to only use 4 of the 8 QOL dimensions measured in the SF36?  This is a major limitation as it takes a validated health survey but only uses half of the dimensions measured with it.  How does this affect the validity of the study?  This needs to be described more clearly in the methods and in the Discussion.

-              Methods line 115 – the authors state that the NSL “was used to consider the physical health status”.  Firstly, please specify that it is the Japanese NSL that is being considered. Secondly, please clarify why this was chosen as the comparator rather than comparing within the patient population (pre-operative vs. post-operative). Thirdly, clarify whether “ to consider the physical health status” means with respect to a national standard, or whether it means to compare as a reference point, or other.

-              Line 147 – define EZR.

Results

-              Please provide the N and % for all values in the Results. Results are often presented as either N alone or % alone.

-              Suggest revising the language “We failed to ask some patients…”.

-              Table 1 requires better formatting to clarify that Age, Smoking Status, Performance Status are all Categories of Characteristics. Either these need to be left shifted with the sub-variables right shifted, or a change in font (e.g. Bold font) should be used to delineate Categories from Variables

-              Suggest adding Performance Status 3-4 and 5 and indicating if these were 0.

-              Figure 1 needs to be reformatted.  Change “Before Op.” to Baseline or Pre-Operative; Legend needs to define the 4 P-QOL in full rather than acronyms; asterix denoting significance should be placed above the bars on the figure itself, specifically above the bar that is significant (e.g. PF or RP). Consider using a curly bracket to denote where the significance applies to all 4 subscales. Add a footnote / legend to the Figure to describe J-NSL and 95% CI, as well as levels of significance and define the acronyms.

-              Line 168 – do you mean “were compared with” as opposed to “were comparable with”?

-              The authors compare the P-QOL scores with the Japanese NSL.  A better analysis would be to compare the P-QOL at various time points with the baseline pre-operative data.  It looks as though PF and RP do not return to baseline even after 1 year.   Another way to consider showing the data would be to group all the PF bars together for the various time points, and all the RP bars together for the various time points, and show more clearly how the individual scores vary over time and whether they return to baseline.  This is described in section 3.2, but I think should be the main focus of describing the Figure, rather than comparing to J- NSL.

-              Table 2 – similar to Table 1, please improve the formatting to more clearly delineate Categories from subvariables. Add % to the N of patients.

-              Table 2 – “Others” – please clarify that this means “ongoing smoking”  or what it refers to.

-              Table 4 – similar to prior, please improve formatting of the Table, at % to N, clarify what “Others” means under Smoking status

-              Table 5 – please change the term “more smoked”, which does not have the meaning that the authors define. Rather than “X or not” please change to X vs. Y

-              As previously mentioned, a major limitation to the results and the study is that the authors do not indicate what proportion of patients required additional therapies (e.g. adjuvant systemic therapy) or other in the post-operative period within 1 year.  This would potentially significantly affect P-QOL.  The authors should include mention of what proportion of patients received surgical treatment alone vs. received additional therapies within 1 year post-operatively.  Additionally, it suggested that the authors revise the analysis to exclude patients that underwent additional therapies in the post-operative period, so as to eliminate the potential confounding factor from the analysis and in order to ensure the analysis pertains to the impact of surgery alone on QOL.

Discussion

-              First sentence needs to be revised for clarity by changing the wording “extend the prognosis”, and by clarifying the main findings of this specific study.

-              The first paragraph of the Discussion should be a summary of the main findings of this study.  In the second paragraph, start comparison with literature.   This will help clarify the main point of the study in paragraph 1 and build the argument in subsequent paragraphs.

-              Please revise the wording “provide sufficient treatment” on line 234-235.

-              Please revise the wording on lines 241-242

-              Suggest using a term other than “great” to describe negative impact on line 250 e.g. significant.

-              One downside to the analysis is that the study did not allow for comparison of patients who were ongoing smokers compared to those that quit within 1 year. In many other countries, surgery is not withheld from patients with smoking addiction, and it would be important to describe the impact of ongoing smoking compared with pre-operative smoking cessation. Furthermore, many guidelines have suggested smoking cessation 8 weeks before surgery noting that smoking cessation very close to the operative period can actually worsen post-operative outcomes. How do the authors reconcile that smokers quit at least 2 weeks before surgery and the potential increased post-operative risk from quitting too close to surgery?

-              A downside to this study that needs to be acknowledged is that it did not compare the impact of radiation therapy or SABR for early stage lung cancer. Radiation therapy is often used with curative intent for patients with comorbidities or lower PS when surgery is either felt to be too high risk, or where patients choose not to undergo surgery over concerns related to QOL.  It would seem that the next step would be to consider a similar evaluation in patients undergoing RT and compare to those undergoing surgery to identify whether there are patient subgroups that experience less reduction in QOL for similar cure rates.

Conclusion

-              1st sentence should be restructured to better summarize the main study results.

-              “solitary life” does not mean the same thing as “living alone”. Please revise for accuracy

-              The authors conclude that surgery had a negative impact on P-QOL of elderly patients with lung cancer, but this conclusion cannot be drawn without considering what proportion of those patients had also undergone systemic therapies as well in the 1 year post-operatively.

Comments on the Quality of English Language

These are embedded within the file attached under Comments and Suggestions for Authors.

Author Response

A major limitation to the study is that the authors do not indicate whether surgical treatment alone was curative, and what proportion of patients required additional therapies (e.g. adjuvant systemic therapy) in the post-operative period within 1 year.  Note is made that 29 patients ultimately had Stage II-III disease and would potentially be eligible for adjuvant therapies. This would potentially significantly affect P-QOL post-operatively.  At the minimum, the authors should include mention of what proportion of patients received surgical treatment alone vs. received additional therapies. 

⇒According to your suggestion, we added the description of what proportion of patients received surgical treatment alone vs. received additional therapies including the content of the treatment.

Additionally, it is suggested that the authors revise the analysis to exclude patients that underwent additional therapies in the post-operative period, so as to eliminate the potential confounding factor from the analysis and in order to ensure the analysis pertains to the impact of surgery alone on QOL. 

⇒We think that postoperative adjuvant therapy is common in lung cancer treatment, therefore, we had better include patients who received adjuvant therapy in this QOL evaluation.

Moreover, of the 23 patients who received additional therapies, 10 patients (43%) had oral uracil-tegafur, which is considered less toxic.

 As it stands, the authors conclude “that surgery had a negative impact on P-QOL of elderly patients with lung cancer” but this conclusion is mired by the fact that patients receiving adjuvant therapies were not excluded.

⇒⇒According to your suggestion, we deleted the word “elderly”.

In this study, the majority of the patients (74%) received surgery alone, and of 25 patients who had Stage II or IIIA, 13 patients (52%) consequently had not received adjuvant therapy, probably because of the effect of surgery. Thus, we think surgery substantially affected the patients.

Another limitation is that this study does not include patients that continued to smoke in the post-operative period, limiting comparison to those that smoked within 1 year vs “others”, which is not clearly described, given that the authors state that those that continued smoking were denied surgery (which is not a universal standard in cancer care).

⇒” others” mean never smokers and the patients who stopped smoking more than 1 year from surgery. We indeed deny surgery for the patients who continue smoking, because it is very important to prevent postoperative pneumonia, which could be lethal. We (surgeons and physicians) instruct patients about smoking cession also during postoperative period continuously. We ask patients and their family whether they resume smoking routinely and repeatedly, I think we have almost none who resume smoking in lung cancer patients after our instruction.

Additionally, the main comparison made in physical QOL scores is to the Japanese national standard level. The more important comparison is the comparison at the various post-operative time periods with the pre-operative baseline, and should be the main Figure of the paper. While this comparison is described in section 3.2, it should be the focus of the analysis.

⇒We compared between baseline and various postoperative time periods in physical P-QOL scores on statistical analysis. According to your indication, we changed the sentence in section 3.2..

Simple Summary:

-              Line 32-33 - Suggest changing “sufficient treatment” to “appropriate treatment” or “curative treatment”

⇒According to your suggestion, we changed the word.

Abstract:

-              Line 42 - Suggest rewording “strongest physical load” to “highest physical risk” or “highest physical burden”.  Consider being more consistent in this terminology throughout the paper.

⇒According to your suggestion, we changed the words.

-             Line 43 - Rather than “prone to be affected by surgery” consider “at higher risk of complications” or “more likely to experience physical decline due to surgery”

⇒According to later your suggestion, we deleted this sentence.

-              Abstract must include details about the number of patients studied and numerical data regarding the results. For the main results stated in the abstract, numerical data with p-values should be presented. 

⇒According to your suggestion, we inserted the number of patients studied and numerical data with p-values.

-              Condense the first 3 sentences of the abstract to allow for additional data to be included.

⇒According to your suggestion, we revised the first 3 sentences.

-              Significant revision of the grammatical structure is required.  E.g. rather than “patient who fulfill the aforementioned conditions”, consider  “patients with these conditions”. There should be no comma after “additional support” on line 53.

⇒According to your suggestion, we revised this sentence.

Introduction

-              Line 59 – 60 - 1st sentence – rather than “for early stage-lung cancer” consider “for patients with early stage lung cancer”

⇒According to your suggestion, we revised this sentence.

-              The 2nd paragraph of the Introduction can be significantly shortened for brevity.

⇒According to your suggestion, we revised the 2nd paragraph.

-              Line 71  - again, the term “sufficient medical treatment” is suboptimal and should be revised to either “optimal treatment” or “curative intent treatment”.

⇒As a result of the revision of the 2nd paragraph, we deleted this sentence.

-              Lines 76-78 need to be clarified. What changes in QOL are associated with survival and what does “progress of treatment” mean?

⇒According to your suggestion, we revised this sentence.

Methods

-              Line 91 - Rather than “Finally”, the better wording is “Ultimately”

⇒According to your suggestion, we changed the word.

-              Please add the percentages for all numbers provided e.g. 100 patients (%); 74 patients (%) undergoing thoracoscopic surgery and % undergoing thoracotomy as N (%).

⇒According to your suggestion, we revised this sentence.

-              In the methods, there needs to be more description about how the authors managed the fact that some patients were not given the survey to complete at certain time points – how did this factor into the evaluation?

⇒According to your suggestion, we added descriptions of the patients who were excluded from this study.

-              Please describe where patients were consented? At clinic? At pre-operative visit? By phone?

⇒According to your suggestion, we revised the description.

-              What does “mode of living” mean? Please define categories. Note is made that this is defined in Section 2.5. Suggest including definitions up front for clarity

⇒I used “mode of living” for categorizing the living conditions of the patients, i.e. living alone or living somebody. I recognized my improper usage of words due to your question. I revised my usage of words, then I chose different words, “living conditions” .  

-              From the study methodology, it’s not clear why “the scores at 6 months after surgery were chosen”. The rest of the sentence makes it sound as though data were  collected at multiple post-operative time points but only the 6 month data were used. The methodology should include a description of the frequency of the post-operative survey data collection. (This is described in Section 2.1. The methods should be reorganized to include this information earlier).

⇒According to your suggestion, we revised the description.

-              Line 106  - Was the described outpatient consultation with the surgeon, anesthesia, or other?

⇒According to your suggestion, we revised the description.

-              More description of the 4 subscales of the SF-36 is required in the methods. What does “role-physical” include? What are the components that make up these subscales? Consider including the SF-36 as an Appendix.

⇒According to your suggestion, we added the appendix for the SF-36.

-              What smoking cessation tools were used at the first visit? 2.3 simply states that patients were instructed to stop smoking. By whom? What supports were put in place? Was any information or tools provided?

⇒According to your suggestion, we added a few sentences.

-              Line 122 - How was it “confirmed adequately whether they stopped smoking completely”? Did a research assistant call? Was it by survey? Were patients asked at a follow-up visit?

⇒According to your suggestion, we revised the description.

-              Line 124-125 – what does “others” mean with respect to smoking cessation.

⇒Sorry, “others” mean “the others”. we revised the description.

-              Line 133 – rather than “such as”, change to “including living alone and living with somebody” which is clearer.

⇒According to your suggestion, we revised the description.

-              The Methods section would benefit from reorganization to ensure clarity.  The 2nd paragraph of the Methods should move to just before the Statistical Analysis as it describes how the variables were used to assess the relationship with physical QOL.

⇒According to your suggestion, we revised the description.

-              When describing performance status, all 4 levels should be described, not just 1-3 and 5.

⇒According to your suggestion, we revised the description.

-              Line 104 – the Methods state that “each patient completed the questionnaire…” but in the results, the authors state that some patients weren’t given the questionnaire.  As such, it would be more clear to simply state “Patients were asked to complete the questionnaire …” and in the results, clarify how many patients were given the questionnaire.

⇒According to your suggestion, we revised the description.

-              Why did the authors choose to only use 4 of the 8 QOL dimensions measured in the SF36?  This is a major limitation as it takes a validated health survey but only uses half of the dimensions measured with it.  How does this affect the validity of the study?  This needs to be described more clearly in the methods and in the Discussion.

⇒According to your indication, we revised the content. We added one more literature cited.

-              Methods line 115 – the authors state that the NSL “was used to consider the physical health status”.  Firstly, please specify that it is the Japanese NSL that is being considered.

⇒In SF-36, NSL is calculated on each age in each country (e.g. 50s, 60s, 70s)

Secondly, please clarify why this was chosen as the comparator rather than comparing within the patient population (pre-operative vs. post-operative).

⇒Our patients (especially in pre-operative period) were asymptomatic, their general conditions were almost normal, therefore, we compared with NSL. We wanted to know the patient status compared with healthy people. We also conducted comparison within the patient population.

Thirdly, clarify whether “ to consider the physical health status” means with respect to a national standard, or whether it means to compare as a reference point, or other.

⇒It means with respect to a national standard level.

-              Line 147 – define EZR.

⇒EZR is the name of free statistical software.

Results

-              Please provide the N and % for all values in the Results. Results are often presented as either N alone or % alone.

⇒According to your suggestion, we revised the description.

-              Suggest revising the language “We failed to ask some patients…”

⇒According to your suggestion, we revised the description.

-              Table 1 requires better formatting to clarify that Age, Smoking Status, Performance Status are all Categories of Characteristics. Either these need to be left shifted with the sub-variables right shifted, or a change in font (e.g. Bold font) should be used to delineate Categories from Variables

⇒According to your suggestion, we revised the description.

-              Suggest adding Performance Status 3-4 and 5 and indicating if these were 0.

⇒According to your suggestion, we revised the description.

Figure 1 needs to be reformatted.  Change “Before Op.” to Baseline or Pre-Operative; Legend needs to define the 4 P-QOL in full rather than acronyms; asterix denoting significance should be placed above the bars on the figure itself, specifically above the bar that is significant (e.g. PF or RP). Consider using a curly bracket to denote where the significance applies to all 4 subscales. Add a footnote / legend to the Figure to describe J-NSL and 95% CI, as well as levels of significance and define the acronyms.

⇒According to your suggestion, we revised the description.

-              Line 168 – do you mean “were compared with” as opposed to “were comparable with”?

⇒I mean “were equal”. I changed the expression.

-              The authors compare the P-QOL scores with the Japanese NSL.  A better analysis would be to compare the P-QOL at various time points with the baseline pre-operative data.  It looks as though PF and RP do not return to baseline even after 1 year.   Another way to consider showing the data would be to group all the PF bars together for the various time points, and all the RP bars together for the various time points, and show more clearly how the individual scores vary over time and whether they return to baseline.  This is described in section 3.2, but I think should be the main focus of describing the Figure, rather than comparing to J- NSL.

⇒We compared the P-QOL at various time points with the baseline preoperative data, not with the Japanese NSL on our statistical analysis. Figure 1 shows the results. I think the perioperative progress is more important than comparing with the Japanese NSL, too. We show P-QOL scores of each subscale at various times as they are. I showed the Japanese NSL for reference values. I think that analysis of the progress of the individual scores is probably interesting, although we did not conduct it in this study.

-              Table 2 – similar to Table 1, please improve the formatting to more clearly delineate Categories from subvariables. Add % to the N of patients.

⇒According to your suggestion, we revised the description.

-              Table 2 – “Others” – please clarify that this means “ongoing smoking”  or what it refers to.

⇒”Others mean the patients who never smoked or stopped smoking more than 1 year before surgery. We changed the description.

-              Table 4 – similar to prior, please improve formatting of the Table, at % to N, clarify what “Others” means under Smoking status

⇒According to your suggestion, we revised the description.

-              Table 5 – please change the term “more smoked”, which does not have the meaning that the authors define. Rather than “X or not” please change to X vs. Y

⇒If we change to X vs.Y, the rationality of the numbers of 95%CI is thrown into disorder, therefore, we use the aforementioned expression (i.e. stopped smoking within 1 year).

-              As previously mentioned, a major limitation to the results and the study is that the authors do not indicate what proportion of patients required additional therapies (e.g. adjuvant systemic therapy) or other in the post-operative period within 1 year.  This would potentially significantly affect P-QOL.  The authors should include mention of what proportion of patients received surgical treatment alone vs. received additional therapies within 1 year post-operatively.  Additionally, it suggested that the authors revise the analysis to exclude patients that underwent additional therapies in the post-operative period, so as to eliminate the potential confounding factor from the analysis and in order to ensure the analysis pertains to the impact of surgery alone on QOL.

⇒According to your suggestion, we added the description regarding adjuvant therapy.

Discussion

-              First sentence needs to be revised for clarity by changing the wording “extend the prognosis”, and by clarifying the main findings of this specific study.

⇒According to your suggestion, we revised the description.

-              The first paragraph of the Discussion should be a summary of the main findings of this study.  In the second paragraph, start comparison with literature.   This will help clarify the main point of the study in paragraph 1 and build the argument in subsequent paragraphs.

⇒According to your suggestion, we revised the description.

-              Please revise the wording “provide sufficient treatment” on line 234-235.

⇒According to your suggestion, we revised the description.

-              Please revise the wording on lines 241-242

⇒According to your suggestion, we revised the description.

-              Suggest using a term other than “great” to describe negative impact on line 250 e.g. significant.

⇒According to your suggestion, we revised the description.

-              One downside to the analysis is that the study did not allow for comparison of patients who were ongoing smokers compared to those that quit within 1 year. In many other countries, surgery is not withheld from patients with smoking addiction, and it would be important to describe the impact of ongoing smoking compared with pre-operative smoking cessation. Furthermore, many guidelines have suggested smoking cessation 8 weeks before surgery noting that smoking cessation very close to the operative period can actually worsen post-operative outcomes. How do the authors reconcile that smokers quit at least 2 weeks before surgery and the potential increased post-operative risk from quitting too close to surgery?

⇒In reinvestigation for the duration of preoperative smoking cession, there were 8 patients who had stopped smoking within 8 weeks before surgery (mean±SD, 23.9±8.7days, range 14-42 days). We think smoking cession more than 4 weeks would be desirable for safe postoperative management, however, its implementation sometimes difficult due to disease state or schedule of patient. We did not experience any major postoperative complication among these patients including pneumonia.

-              A downside to this study that needs to be acknowledged is that it did not compare the impact of radiation therapy or SABR for early stage lung cancer. Radiation therapy is often used with curative intent for patients with comorbidities or lower PS when surgery is either felt to be too high risk, or where patients choose not to undergo surgery over concerns related to QOL.  It would seem that the next step would be to consider a similar evaluation in patients undergoing RT and compare to those undergoing surgery to identify whether there are patient subgroups that experience less reduction in QOL for similar cure rates.

⇒Thank you for your suggestion. We agree your opinion, therefore, in the same duration we conducted radiation therapy or SABR for certain early stage lung cancer cases as you mention. Our purpose was to understand how surgery impacted the patients at the level of daily life.

Conclusion

-              1st sentence should be restructured to better summarize the main study results.

⇒According to your suggestion, we revised the description.

-              “solitary life” does not mean the same thing as “living alone”. Please revise for accuracy

⇒According to your suggestion, we revised the description.

-              The authors conclude that surgery had a negative impact on P-QOL of elderly patients with lung cancer, but this conclusion cannot be drawn without considering what proportion of those patients had also undergone systemic therapies as well in the 1 year post-operatively.

⇒According to your suggestion, we revised the description.

Reviewer 2 Report

Comments and Suggestions for Authors

Check up the spacing and hyphen between ‘non-small cell lung cancer.

Remove the hyphen between stage and lung. ‘early stage lung cancer.’

The expression of ‘bodily pain’ is unusual. I think that ‘body pain’ will be fine.

In the statistical analysis, what do you mean by ‘multiple regression analysis’? Do you mean multiple analyses of the univariate regression models? Would you clarify the details of the statistical method?

In the tables, please separate the titles and the items of the categories. The current state is not readable.

In table 1, ‘p-stage’ is better to change to ‘pathologic stage’ or ‘TNM stage’.

In general, three decimal places are common to report p-value.

In table 5, what does ‘estimate’ mean? And, the negative value of ‘estimate’ is correct? Please check up and clarify enough that readers easily understand.    

Comments on the Quality of English Language

It needs minor revision in English

Author Response

Check up the spacing and hyphen between ‘non-small cell lung cancer.

⇒According to your suggestion, we revised the description.

Remove the hyphen between stage and lung. ‘early stage lung cancer.’

⇒According to your suggestion, we revised the description.

The expression of ‘bodily pain’ is unusual. I think that ‘body pain’ will be fine.

⇒This expression is commonly used in the studies using SF-36.

In the statistical analysis, what do you mean by ‘multiple regression analysis’? Do you mean multiple analyses of the univariate regression models? Would you clarify the details of the statistical method?

⇒We conducted multivariate analysis (multiple regression amalysis) for the factors which showed statistically significant difference on univariate analysis. According to your suggestion, we revised the description.

In the tables, please separate the titles and the items of the categories. The current state is not readable.

⇒According to your suggestion, we revised the description.

In table 1, ‘p-stage’ is better to change to ‘pathologic stage’ or ‘TNM stage’.

⇒According to your suggestion, we revised the description.

In general, three decimal places are common to report p-value

⇒According to your suggestion, we revised the description.

In table 5, what does ‘estimate’ mean? And, the negative value of ‘estimate’ is correct? Please check up and clarify enough that readers easily understand.  

⇒According to your suggestion, we revised the description.

Reviewer 3 Report

Comments and Suggestions for Authors

i suggest to clarify in addition to the comments inside the manuscript the focus on elderly

it seems that they are the target but the support and rationale are not evident

Author Response

i suggest to clarify in addition to the comments inside the manuscript the focus on elderly

it seems that they are the target but the support and rationale are not evident

⇒According to your suggestion, we revised the description.

Round 2

Reviewer 1 Report

Comments and Suggestions for Authors

Thank you for the opportunity to re-review this paper.

The authors made some attempts to address the reviewer comments, but they were incompletely addressed, and some were neglected. When changes were made, it wasn’t always clear where these occurred in the manuscript as the authors did not include the revised statements in the response to reviewers document. Meanwhile, there are several sentences that lack clarity due to the suboptimal English grammar and word choice, which distracts and detracts from the paper.   Overall, the authors did not sufficiently address the comments and there are several improvements still required.

While the study is of potential relevance and warrants publication, but several changes are required:

o   First, the authors need to better acknowledge the confounding effects of post-operative adjuvant therapies on P-QOL.  The discussion needs to include a better description of this confounding effect on P-QOL and conclusion should be revised to clarify this as well.

o   Secondly, the results section requires that the data be more clearly laid out in the text, not just in the Tables.

o   Finally, the authors must revise the paper for English language proficiency.  

Specific Comments:

-              The authors should mention in the methods that they collected data regarding post-operative adjuvant systemic therapies and how this data were collected (e.g. post-operative chart review).

-              The Results section needs to start with the total number of patients included in the study first, then describe survey completion rate.

-              Line 213 – change to underwent surgery alone, not “underwent only surgery”; change “whereas” to “while”

-              Line 216 add a % to 11 patients (%), and 1 (%)

-              Uracil-tegafur is not a commonly used systemic therapy regimen in other parts of the world e.g. North America.  The discussion should include mention of the relative toxicities of the adjuvant therapies used and the extent to which these could have potentially affected P-QOL. 

-              Reviewer comment: the authors conclude “that surgery had a negative impact on P-QOL of elderly patients with lung cancer” but this conclusion is mired by the fact that patients receiving adjuvant therapies were not excluded; Response: According to your suggestion, we deleted the word “elderly”.

o   The authors misunderstood the comment.  The issue is not with inclusion of the word elderly (though the statement is improved by removing the word), but that the authors are drawing a conclusion that surgery had a negative impact on P-QOL when patients undergoing adjuvant therapies were included.  Rather, it would be more accurate to state that “Patients with lung cancer experience a reduced P-QOL in the post-operative period, which may be due to the effects of surgery and other adjuvant therapies”.  While surgery likely had a significant impact, surgery was not the only intervention patients received in the post-operative period, so either the authors need to exclude patients who underwent adjuvant therapies if they want to make this conclusion, or acknowledge that other therapies likely contributed to the effect. The authors can state make the argument that, because most patients underwent surgery alone, the evaluation predominantly reflects the effects of surgery on P-QOL, but the potential contribution of adjuvant therapies to P-QOL cannot be ignored and should be better clarified.  There should be mention that future studies can consider trying to better differentiate the effects of surgery from the effects of adjuvant therapies on P-QOL and their potential co-relationships.

-              Please change the terms “others” and “less smokers” to “never or remote smokers” throughout the manuscript to clarify the language.  The authors need to add a section in the Discussion stating that the findings may not be generalizable as not all centers/countries deny surgery to patients who continue to smoke, thus smoking cessation in the post-operative period is a counfounding variable.   

-              While the authors are correct that smoking cessation reduced post-operative cardiopulmonary complications, it has also been shown that smoking cessation less than 8 weeks prior to surgery can actually increase the risk of post-operative complications such as pneumonia. The authors need to explicitly mention this.

-              The authors were asked to clarify what “progress of treatment” meant.  They changed this to “treatment progress”, which does not help clarify the meaning of the sentence.

-              Stating that patients were consented “during hospital visits as an outpatient” does not clarify where patients were consented – which outpatient visits, with which physicians?  Are these pre-operative clinic visits at which patients are being consented for surgery? Or are these pre-operative anesthesia assessments?

-              Other comments:

o   Simple summary:

§  “Surgery is the most effective for early-stage lung cancer”. The word “treatment” is missing after the word effective.

§  This sentence would also be better revised to state that it poses a heavy physical burden rather than the “heaviest physical burden among the treatment options”

§  Rather than “at our institution”, specify the name and location of the institution

§  The phrase “adequate treatment” is once again features. It had been suggested to the authors to rephrase this through the manuscript

o   Abstract  - surgery poses a high physical burden, but may not necessarily always pose the highest of all treatment options. Please revise.

o   It had been suggested to the authors that the manuscript be revised for expertise in English.  There are still several sentences that are suboptimally written and for which the authors are strongly suggested to have the manuscript reviewed by an English-speaking expert to ensure the intended meaning throughout.

o   Line 68 – change prospectively investigated to prospectively measured

o   The p values are included, but the actual data need to be included as well

-              Line 116 – change to “100 patients provided written informed consent to participate in the study during the pre-operative visit” and specify whether this is “at the time of being consented for surgery”

-              Line 117-118 – not clear, please revise. How long pre-operatively?  What is the time interval between the survey and surgery?  What were the post-operative time frames for survey completion?  Were these at post-operative visits with the surgeon?

-              Line 124 – “n = 2, cerebral infarction and severe brain edema each” suggests that 2 patients had each of these complications.  Was it 1 each?

-              Line 132-133 is not clear.  What does “bodily functions” mean?  Please clarify which 4 domains were selected here, and why.  Lines 132-134 should probably move to section 2.1

-              Section 2.2 requires revision for better clarity of language.  “less smoked” is not a clear term. Consider “never or remote smokers” as a more clear term.

-              The term EZR is still not clarified.  The authors state this is a statistical software. This needs to be mentioned, with the trademark identified and if EZR represents a short-form, this also needs to be clarified.

-              Line 208 – the response rate should be the number of patients that completed the survey.  The percentage completion of the surveys reflects the number of omitted responses (not blanks spaces).

-              Table 1 – line 235. Please change the term LS to Remote or never smokers.  (same with all the other figures)

-              Figure 1 – the legend should be featured underneath the Figure. Black background should be removed.  

-              Results 3.3 – it is not sufficient to only include p values in the text. The raw data needs to be presented in text as well (X vs Y, p = XXX).

-              “at Shonan Kamakura General Hospital or other participating hospitals” – what were the other participating hospitals? Are they all tertiary care or are they community referring centres”? Please describe the location of the study in more detail.  It would also be helpful to know how many thoracic surgeries are performed yearly at this center to better understand the context

-              Line 289 “Preoperatively, the patients lived alone”, missing the word “who” before “lived alone”.

-              Line 390 – significantly lower, not significant lower

-              Line 473 – “whereas” does not have the intended meaning here.  “while” is probably a better word

-              Line 389 – do the authors mean “even though the majority of surgeries were performed thoracoscopically”.

-              Discussion – rather than restate the purpose of the study in the 2nd sentence, summary the main results e.g. In this study, we found that ……

-              Line 421 – “More attention from nurses and social workers is also needed for their support”.  The authors are making inferences about how to mitigate the effects of living alone.  The language should be clarified to more clearly state that living alone is a factor that needs to be considered in terms of risks to P-QOL, and that mitigating strategies need to be explored, but it would be premature to state that “higher awareness” and “more attention” will change outcomes.

-              Discussion – smoking cessation.  The authors should acknowledge that, despite all patients having quite smoking prior to surgery, the potential HR-QOL benefits that accompany smoking cessation did not overcome the negative P-QOL effects of surgery and associated adjuvant therapies.

-              Line 468 – perhaps more accurate to state that the SF-36 can potentially identify patient factors that increase the likelihood of lower post-operative P-QOL.

Comments on the Quality of English Language

Please see above. 

Author Response

o   First, the authors need to better acknowledge the confounding effects of post-operative adjuvant therapies on P-QOL.  The discussion needs to include a better description of this confounding effect on P-QOL and conclusion should be revised to clarify this as well.

⇒We added the description in the discussion according to your suggestion.

o   Secondly, the results section requires that the data be more clearly laid out in the text, not just in the Tables.

⇒We revised the description according to your suggestion.

o   Finally, the authors must revise the paper for English language proficiency.  

⇒We requested a special firm to do English proofreading thoroughly.

Specific Comments:

-              The authors should mention in the methods that they collected data regarding post-operative adjuvant systemic therapies and how this data were collected (e.g. post-operative chart review).

⇒We added the description in the method.

-              The Results section needs to start with the total number of patients included in the study first, then describe survey completion rate.

⇒We added the description according to your comments.

-              Line 213 – change to underwent surgery alone, not “underwent only surgery”; change “whereas” to “while”

⇒We changed the description according to your comments.

-              Line 216 add a % to 11 patients (%), and 1 (%)

⇒We added the description according to your comments.

Uracil-tegafur is not a commonly used systemic therapy regimen in other parts of the world e.g. North America.  The discussion should include mention of the relative toxicities of the adjuvant therapies used and the extent to which these could have potentially affected P-QOL. 

⇒We revised the description according to your suggestion.

-              Reviewer comment: the authors conclude “that surgery had a negative impact on P-QOL of elderly patients with lung cancer” but this conclusion is mired by the fact that patients receiving adjuvant therapies were not excluded; Response: According to your suggestion, we deleted the word “elderly”.

o   The authors misunderstood the comment.  The issue is not with inclusion of the word elderly (though the statement is improved by removing the word), but that the authors are drawing a conclusion that surgery had a negative impact on P-QOL when patients undergoing adjuvant therapies were included.  Rather, it would be more accurate to state that “Patients with lung cancer experience a reduced P-QOL in the post-operative period, which may be due to the effects of surgery and other adjuvant therapies”.  While surgery likely had a significant impact, surgery was not the only intervention patients received in the post-operative period, so either the authors need to exclude patients who underwent adjuvant therapies if they want to make this conclusion, or acknowledge that other therapies likely contributed to the effect. The authors can state make the argument that, because most patients underwent surgery alone, the evaluation predominantly reflects the effects of surgery on P-QOL, but the potential contribution of adjuvant therapies to P-QOL cannot be ignored and should be better clarified.  There should be mention that future studies can consider trying to better differentiate the effects of surgery from the effects of adjuvant therapies on P-QOL and their potential co-relationships.

⇒Thank you for your kind advice and suggestion. We added the description in the discussion.

-              Please change the terms “others” and “less smokers” to “never or remote smokers” throughout the manuscript to clarify the language.  The authors need to add a section in the Discussion stating that the findings may not be generalizable as not all centers/countries deny surgery to patients who continue to smoke, thus smoking cessation in the post-operative period is a counfounding variable.   

⇒We revised the description according to your suggestion.

-              While the authors are correct that smoking cessation reduced post-operative cardiopulmonary complications, it has also been shown that smoking cessation less than 8 weeks prior to surgery can actually increase the risk of post-operative complications such as pneumonia. The authors need to explicitly mention this.

⇒We checked literatures on the guidelines for enhanced recovery after lung surgery including smoking cession, then we added the description.

-              The authors were asked to clarify what “progress of treatment” meant.  They changed this to “treatment progress”, which does not help clarify the meaning of the sentence.

⇒We revised the description according to your comment.

-              Stating that patients were consented “during hospital visits as an outpatient” does not clarify where patients were consented – which outpatient visits, with which physicians?  Are these pre-operative clinic visits at which patients are being consented for surgery? Or are these pre-operative anesthesia assessments?

⇒We added the description.

-              Other comments:

o   Simple summary:

  • “Surgery is the most effective for early-stage lung cancer”. The word “treatment” is missing after the word effective.

⇒We added the word “treatment”.

  • This sentence would also be better revised to state that it poses a heavy physical burden rather than the “heaviest physical burden among the treatment options”

⇒We revised the description.

  • Rather than “at our institution”, specify the name and location of the institution

⇒We changed the description.

  • The phrase “adequate treatment” is once again features. It had been suggested to the authors to rephrase this through the manuscript

⇒We changed the description.

o   Abstract  - surgery poses a high physical burden, but may not necessarily always pose the highest of all treatment options. Please revise.

⇒We revised the description according to your suggestion.

o   It had been suggested to the authors that the manuscript be revised for expertise in English.  There are still several sentences that are suboptimally written and for which the authors are strongly suggested to have the manuscript reviewed by an English-speaking expert to ensure the intended meaning throughout.

⇒We requested a special firm to do English proofreading thoroughly.

o   Line 68 – change prospectively investigated to prospectively measured

⇒We changed the description according to your suggestion.

o   The p values are included, but the actual data need to be included as well

⇒We added the description according to your suggestion.

-              Line 116 – change to “100 patients provided written informed consent to participate in the study during the pre-operative visit” and specify whether this is “at the time of being consented for surgery”

⇒We revised the description according to your suggestion.

-              Line 117-118 – not clear, please revise. How long pre-operatively?  What is the time interval between the survey and surgery?  What were the post-operative time frames for survey completion?  Were these at post-operative visits with the surgeon?

⇒We revised the description according to your suggestion.

-              Line 124 – “n = 2, cerebral infarction and severe brain edema each” suggests that 2 patients had each of these complications.  Was it 1 each?

⇒We meant it was 1 each. One patient had cerebral infarction, and the other had brain edema. We revised the description.

Line 132-133 is not clear.  What does “bodily functions” mean?  Please clarify which 4 domains were selected here, and why.  Lines 132-134 should probably move to section 2.1

⇒We revised the description. Please check.

-              Section 2.2 requires revision for better clarity of language.  “less smoked” is not a clear term. Consider “never or remote smokers” as a more clear term.

⇒We revised the description according to your suggestion.

-              The term EZR is still not clarified.  The authors state this is a statistical software. This needs to be mentioned, with the trademark identified and if EZR represents a short-form, this also needs to be clarified.

⇒EZR is a free statistical software, which was modified the R commander by the department of hematology, Jichi Medical University Saitama Medical Center. We added the description. We could not find the short-form of EZR. EZR had been cited in more than 10,000 English articles until December, 2023. I hope our description is enough. URL of EZR is as follows: 無料統計ソフトEZR (Easy R) (jichi.ac.jp)

-              Line 208 – the response rate should be the number of patients that completed the survey.  The percentage completion of the surveys reflects the number of omitted responses (not blanks spaces).

⇒Thank you for your comments. We revised the description.

-              Table 1 – line 235. Please change the term LS to Remote or never smokers.  (same with all the other figures)

⇒Thank you for your comments. We changed the term.

-              Figure 1 – the legend should be featured underneath the Figure. Black background should be removed.  

⇒We revised the description.

-              Results 3.3 – it is not sufficient to only include p values in the text. The raw data needs to be presented in text as well (X vs Y, p = XXX).

⇒We revised the description.

-              “at Shonan Kamakura General Hospital or other participating hospitals” – what were the other participating hospitals? Are they all tertiary care or are they community referring centres”? Please describe the location of the study in more detail.  It would also be helpful to know how many thoracic surgeries are performed yearly at this center to better understand the context

⇒Thank you for your comments. We added the description including our job situation.

-              Line 289 “Preoperatively, the patients lived alone”, missing the word “who” before “lived alone”.

⇒We revised the description.

-              Line 390 – significantly lower, not significant lower

⇒We revised the description. Thank you.

-              Line 473 – “whereas” does not have the intended meaning here.  “while” is probably a better word

⇒We revised the description.

-              Line 389 – do the authors mean “even though the majority of surgeries were performed thoracoscopically”.

⇒We revised the description.

-              Discussion – rather than restate the purpose of the study in the 2nd sentence, summary the main results e.g. In this study, we found that ……

⇒Thank you for your comments. We revised the description.

-              Line 421 – “More attention from nurses and social workers is also needed for their support”.  The authors are making inferences about how to mitigate the effects of living alone.  The language should be clarified to more clearly state that living alone is a factor that needs to be considered in terms of risks to P-QOL, and that mitigating strategies need to be explored, but it would be premature to state that “higher awareness” and “more attention” will change outcomes.

⇒Thank you for your comments. Yes, we think living alone is a factor that needs to be considered in terms of risks to P-QOL, however, we don’t think “higher awareness” and “more attention” will change outcomes. I think we have to be sensitive to living conditions of patients during perioperative period to avoid deteriorating their QOL. We revised the description.

-              Discussion – smoking cessation.  The authors should acknowledge that, despite all patients having quite smoking prior to surgery, the potential HR-QOL benefits that accompany smoking cessation did not overcome the negative P-QOL effects of surgery and associated adjuvant therapies.

⇒Yes, you are right. However, we can’t change a fact that a patient smoked for many years before surgery. We also acknowledge the potential HR-QOL benefits that accompany smoking cession did not overcome the negative P-QOL effects of surgery and adjuvant therapies. Under this recognition, we intended to consider minimizing the negative effect of surgery and adjuvant therapies in discussion.

-              Line 468 – perhaps more accurate to state that the SF-36 can potentially identify patient factors that increase the likelihood of lower post-operative P-QOL.

⇒Thank you for your comments. We revised the description.

Reviewer 3 Report

Comments and Suggestions for Authors I support all the changes but I added two comments that were not considered by the authors.

Author Response

evidence is required for this statement(Page 5)

⇒We added a reference.

why? what is the rationale for this

⇒We added the description to clarify our purpose of this study.